# Coordinated adaptations define the ontogenetic shift from worm- to fish-hunting in a venomous cone snail

Aymeric Rogalski ®[1], S. W. A. Himaya[1] & Richard J. Lewis ®[1] ✉

Marine cone snails have attracted researchers from all disciplines but early life stages have received limited attention due to difficulties accessing or rearing juvenile specimens. Here, we document the culture of *Conus magus* from eggs through metamorphosis to reveal dramatic shifts in predatory feeding behaviour between post-metamorphic juveniles and adult specimens. Adult *C. magus* capture fish using a set of paralytic venom peptides combined with a hooked radular tooth used to tether envenomed fish. In contrast, early juveniles feed exclusively on polychaete worms using a unique "sting-and-stalk" foraging behaviour facilitated by short, unbarbed radular teeth and a distinct venom repertoire that induces hypoactivity in prey. Our results demonstrate how coordinated morphological, behavioural and molecular changes facilitate the shift from worm- to fish-hunting in *C. magus*, and showcase juvenile cone snails as a rich and unexplored source of novel venom peptides for ecological, evolutionary and biodiscovery studies.

Throughout the history of life, evolutionary innovations have allowed evolving lineages to acquire new functions that open up ecological opportunities and, in many cases, promote diversification[1,2]. Understanding how these transitions have occurred can be challenging, with observed traits often arising from a series of evolutionary changes that eventually culminate into a complex trait[3,4]. The venom apparatus of marine cone snails (Gastropoda: Conidae) is an example of evolutionary innovation that evolved through morphological modifications of the foregut[5], promoting the extensive radiation of the group since the Eocene, with over 1000 extant species distributed worldwide[6]. This group of predatory gastropods has evolved within a biphasic lifecycle, with most species hatching as free-swimming larvae that become benthic carnivorous juveniles after metamorphosis[7,8]. Predatory feeding after metamorphosis relies on the deployment of potent neurotoxins (conotoxins) secreted in a long tubular venom gland and injected via highly modified, hollow radular teeth[9,10]. This sophisticated feeding strategy has allowed these slow-moving predators to initially feed on worms, and more recently facilitated the evolutionary shift to mollusc- and fish-hunting[11,12].

Because of their recent and extensive radiation and the plethora of venom peptides they produce, cone snails have attracted interest from evolutionary biologists[11], pharmacologists[13] and toxicologists[14], but this broad interest contrasts with the scarcity of literature on early life stages. Observations of juveniles in the field have been hindered by their minute size and their identification often limited by high morphological similarity between related species[15–17]. On the other hand, challenges rearing cone snails have restricted previous investigations to the exploration of embryonic and larval stages[18–21]. Because of these limitations, the ecology and biochemistry of juvenile cone snails have been largely overlooked. This extends to widely-studied species such as the Magician's cone (*Conus magus* Linnaeus, 1758), the source of the FDA-approved analgesic Prialt® (ω-conotoxin MVIIA)[22]. Based on dissected wild-caught specimens, *C. magus* was suggested to undergo a dietary shift from worm- to fish-hunting during ontogeny[23], but empirical evidence is lacking due to challenges accessing early life stages.

Here we cultured *Conus magus* from egg capsules to hatching larvae, and through metamorphosis to carnivorous juveniles. Following metamorphosis, juvenile *C. magus* were observed to prey exclusively on polychaete worms using ancestral-like radular teeth and a unique venom repertoire, before switching to fish-hunting in adulthood. Through a combination of experimental approaches, we

[1]Institute for Molecular Bioscience, The University of Queensland, Brisbane 4072 QLD, Australia. ✉e-mail: r.lewis@uq.edu.au

demonstrate how the transition from worm- to fish-hunting during ontogeny is marked by a series of coordinated changes that span all levels of biological organisation. Our results show how laboratory-reared specimens can provide new insights into the ecology of secretive life stages, and highlight the potential of juvenile cone snails as an untapped source of novel bioactive venom peptides that would otherwise only be accessible through exon capture or genome sequencing.

## Results and discussion
### Larval development and venom apparatus morphogenesis
Like the vast majority of caenogastropods, cone snails are gonochoric (males and females exist as separate individuals), with female *C. magus* depositing their eggs in semi-rigid egg capsules, usually on the underside of a coral rock (Fig. 1a, b; Supplementary Movie 1). Based on egg diameter, this species was previously thought to undergo metamorphosis immediately after hatching (lecithotrophy)[24], but instead we found that the larvae had an obligatory free-swimming, filter-feeding phase (planktotrophy), consistent with the broad distribution of this species across the Indo-Pacific region[25]. After 21 days, the egg capsules released planktonic larvae that had a thin, semi-transparent shell consisting of 1.5 whorls and measuring $712 \pm 30\,\mu m$ (mean $\pm$ SD, $n = 10$) in length. Swimming and feeding in larval stages was facilitated by the velum, which was divided into four lobes. Each lobe had two bands of ciliated cells running along its periphery, and yellow pigmented cells distributed along its margins (Fig. 1c). At 1 day post-hatching (dph), the larval oesophagus consisted of a simple tube of ciliated epithelium that funnelled ingested microalgae from the mouth to the stomach. At ~50 μm from the mouth, a patch of enlarged cells lacking apical cilia was found embedded in the ventral wall of the oesophagus (Fig. 2a; Supplementary Fig. 1a, b). In the subsequent days, this zone of non-ciliated cells extended anteriorly and posteriorly, and ultimately pinched-off from the oesophagus to give rise to the buccal cavity and radular sac, consistent with previous observations on the larva of *Conus lividus*[5].

By the end of larval development (15 dph), the shell had completed 2.5 whorls and measured $1360 \pm 60\,\mu m$ ($n = 10$) in length. At this stage, the foot showed increased flexibility and mobility, and the larvae were now capable of crawling on solid surfaces. The radular sac had originated as an out-pocketing of the buccal cavity, and both structures could be seen as interconnected chambers beneath the oesophagus (Fig. 2b; Supplementary Fig. 1c). Immediately posterior to the radular sac, the prospective venom gland (VG) was evident as a hypertrophied zone of secretory epithelial cells extending down the ventral wall of the oesophagus (Supplementary Fig. 1d). Transmission electron microscopy (TEM) revealed that small, spherical secretory granules started accumulating in the epithelial cell layer of the VG, which remained confluent with the oesophagus until metamorphosis. These results were consistent with observations on *C. lividus*, which showed that most elements of the venom apparatus started to differentiate during larval development[5] to facilitate the rapid switch to carnivory after metamorphosis.

### Metamorphosis marked the transition from herbivory to carnivory
Metamorphosis involved a number of behavioural and structural changes allowing the transition from swimming, filter-feeding larva to benthic, carnivorous juvenile. The process was elicited by the addition of a suitable substrate to induce larval settlement [crustose coralline algae (CCA)-covered rocks]. Following settlement, metamorphosis was initiated by the resorption of the velum into the head region, with the yellow pigments bordering its margin concentrating into two masses on each side of the head (Fig. 1d). In the process, the two bands of ciliated cells were discarded onto the adjacent substrate.

At 1 day post-settlement (dps), the two masses of involuted velar cells arising from the resorption of the velum could be seen on each side of the head, beneath the cerebral ganglia, and the larval mouth

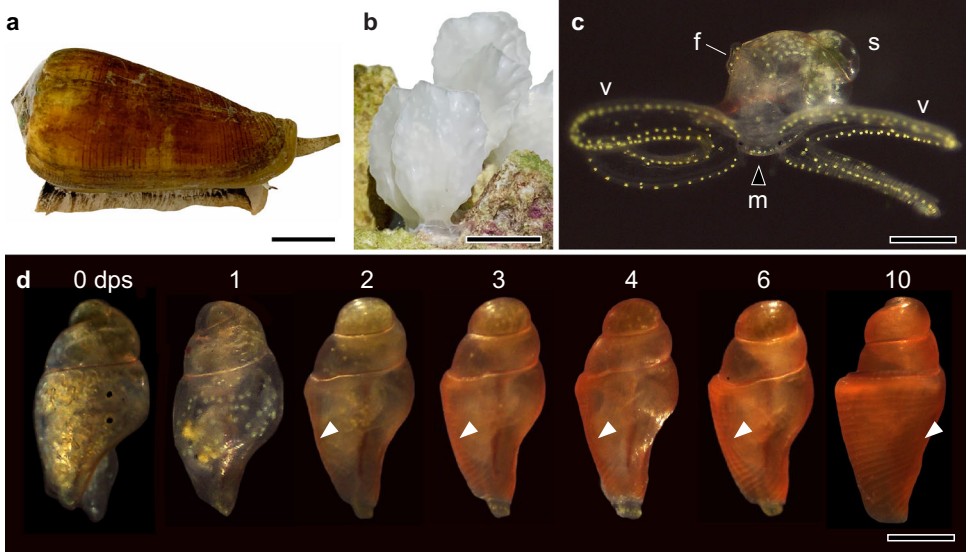

**Fig. 1 | Developmental stages in the biphasic life history of *C. magus*.** *Conus magus* has a biphasic life history that includes planktonic larvae that become benthic carnivorous juveniles after metamorphosis. **a** Adult laying female *C. magus* used in this study. Scale bar = 20 mm. **b** Eggs were deposited in semi-rigid egg capsules, usually laid on the underside of a coral rock. Each capsule contained between 500–700 eggs. Scale bar = 20 mm. **c** After 21 days, the egg capsules released planktonic larvae that had a thin, translucent shell (s) and a ventral foot (f). Swimming and feeding in larval stages was facilitated by the velum (v), which directed captured microalgae to the mouth (m). Scale bar = 0.5 mm.

**d** Metamorphosis and shell calcification in the early juvenile from 0–10 dps (days post-settlement). Settled late larva at 0 dps (=15 dph) showing the velar lobes retracted inside the shell, suggesting imminent metamorphosis. At 1 dps, the velum was completely resorbed into the head region and the yellow pigments concentrated into two masses on each side of the head. In the following days, the shell calcified rapidly, turning bright orange from 2–6 dps. Juveniles were observed to prey on worms from 10 dps, but rapid shell growth after 6 dps suggests carnivory may have started earlier. White arrowheads point to the larval-adult shell boundary. Scale bar = 0.5 mm.

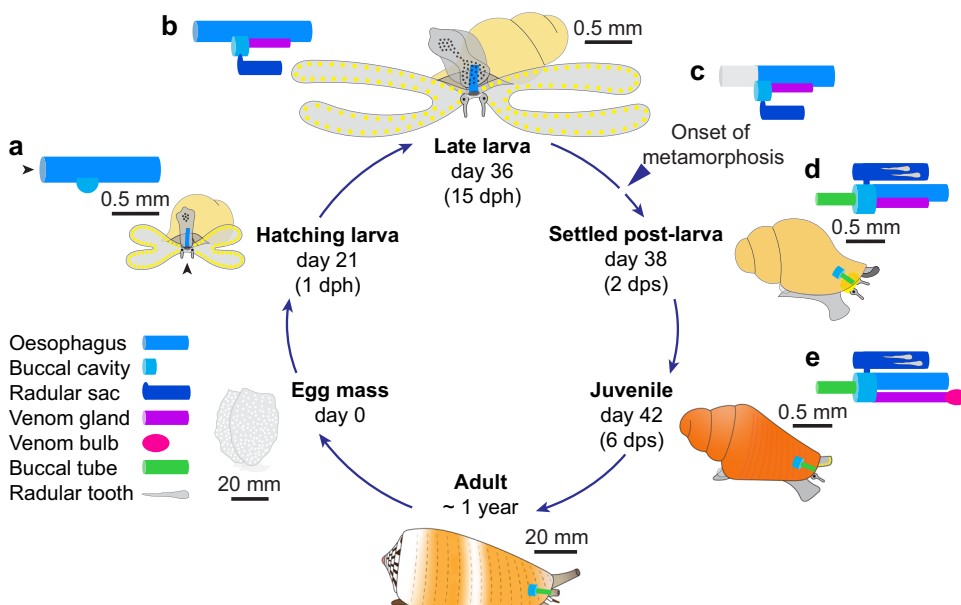

**Fig. 2 | *C. magus* life history and venom apparatus morphogenesis.** Morphogenesis of the venom apparatus was initiated early during larval development and its components elaborated from the distal oesophagus (foregut). **a** The oesophagus of hatching larvae consisted of a simple tube of ciliated epithelium, with the prospective buccal cavity and radular sac rising from a patch of enlarged cells embedded in the ventral wall of the oesophagus (see Supplementary Fig. 1a, b). Black arrowhead indicates position of the mouth. **b** In late larvae, the buccal cavity and radular sac could be seen as interconnected chambers beneath the oesophagus (see Supplementary Fig. 1c). The prospective venom gland was evident as an accumulation of secretory cells in the ventral wall of the oesophagus, posterior to the buccal cavity (see Supplementary Fig. 1d). **c** The onset of metamorphosis was marked by the resorption of the velum and loss of the larval mouth and anterior oesophagus (see Supplementary Fig. 1e). **d** At 2 dps, the buccal tube had formed and chitinous radular teeth started accumulating in the radular sac (see Supplementary Fig. 1f). **e** By 6 dps, the venom gland had pinched-off from the oesophagus, remaining only connected to the buccal cavity, and its distal end became encapsulated with the venom bulb. dph days post-hatching, dps days post-settlement.

and anterior oesophagus were lost (Fig. 2c; Supplementary Fig. 1e). At 2 dps, most involuted velar cells had been cleared from the head region and odontoblasts located in the long arm of the radular sac had begun secreting chitinous radular teeth (Fig. 2d). The proboscis and its sheath (rostrum) had formed, although mitotic profiles of dividing cells indicated ongoing differentiation (Supplementary Fig. 1f). The post-metamorphic mouth was created de-novo at the anterior end of the proboscis, as well as the buccal tube, connecting the mouth to the buccal cavity (Fig. 2d). The rapid apparition of these structures suggests they may arise through transdifferentiation of velar cells, as previously proposed[26]. This is consistent with the concomitant resorption of the velum, and the observation of velar pigmented cells concentrating into the head region and migrating along the oesophagus from 1–3 dps (Fig. 1d).

At 6 dps, all elements of the venom apparatus were present and fully differentiated (Figs. 2e and 3). The short arm of the radular sac had started accumulating mature, chitinous radular teeth and the buccal cavity had narrowed to form the oesophagus, immediately posterior to the radular sac (Fig. 3a, b). At this stage, the VG had detached from the ventral wall of the oesophagus and only remained connected to the buccal cavity (proximal end) (Fig. 3a, c), while its blind end had differentiated into a conspicuous venom bulb (distal end) comprising two layers of muscle fibres enclosing a wide lumen (Fig. 3a, d). The connection between the VG and the bulb was maintained by a thin canal running through the muscular sheath into the lumen of the bulb. The juvenile VG consisted of a tube of secretory cuboidal epithelium surrounded by a thin layer of muscle fibres. The cytoplasm of the secretory cells was filled with a variety of dense, spherical granules, reminiscent of the granules found in injected venoms. The dissociation of the secretory cells from the basal lamina and rupture of their membrane indicate that granules are released through a holocrine process (Fig. 3e; Supplementary Fig. 2b, c). The transition to a benthic lifestyle was marked by the calcification of the shell, which turned a bright orange between 1–6 dps (Fig. 1d), enhancing crypsis after settlement on CCA-covered rocks.

## Prey preference and foraging behaviour in juvenile and adult *C. magus*

The transition to adulthood in *C. magus* was marked by a drastic shift in prey preference accompanied by profound morphological and behavioural changes. While adults are strict piscivores, *Danio rerio* (zebrafish) larvae failed to trigger predatory behaviours of juveniles, which were instead observed to prey exclusively on polychaete worms (family Syllidae) using a "sting-and-stalk" foraging behaviour (Fig. 4a). Early vermivory in *C. magus* was previously inferred from dissected wild-caught specimens, although worm setae were only retrieved in the digestive tracts of three juveniles >4 mm[23]. Additionally, the methods used for the identification of small specimens are not mentioned and the high morphological similarity between juvenile cone snails suggests the sampling could have included other species. The present study provides empirical evidence of strict vermivory in juvenile *C. magus*. The feeding behaviour of juveniles was initiated by extension of the proboscis which probed the surface of the worm in preparation for venom injection. After several minutes, a radular tooth held at the tip of the proboscis was stabbed into the worm and the proboscis rapidly withdrawn inside the rostrum, leaving the prey untethered. Envenomation induced hypoactivity in worm prey, characterised by the loss of normal swimming, hiding and escape behaviours. The snail then stalked its prey for several minutes before extending its rostrum and engulfing the worm whole (Supplementary Movie 2). Occasionally, worms were stung a second time. The same feeding sequence was observed in all juveniles from 10 dps, although histology and rapid shell growth between 6–10 dps suggest carnivory may have started earlier (Fig. 1d). This "sting-and-stalk" foraging behaviour was consistent with the juvenile radular tooth lacking apical barbs, blades and serrations (Fig. 4b; Supplementary Fig. 2a), as seen in

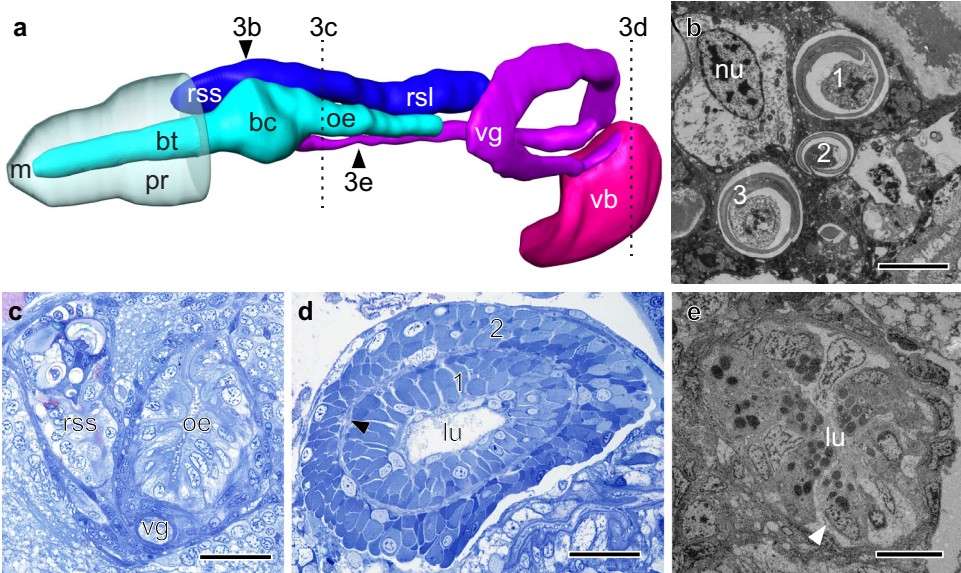

**Fig. 3 | Anatomy of juvenile *C. magus* venom apparatus.** Six days after the onset of metamorphosis, all components of the venom apparatus were present and fully differentiated. **a** Left lateral view of the venom apparatus reconstructed from stained histological sections. Doted lines indicate planes of section shown in **c**, **d**, with the black arrowheads indicating the regions of the radular sac and venom gland sectioned in **b**, **e**, respectively. **b** Transmission electron microscopy (TEM) revealed that the short arm of the radular sac (rss) had started accumulating chitinous radular teeth (numbered 1–3) (see Supplementary Fig. 2a). Scale bar = 5 μm. **c** Just posterior to the buccal cavity, the venom gland (vg) runs beneath the oesophagus (oe). Scale bar = 30 μm. **d** The blind end of the venom gland was encapsulated with the venom bulb (vb), which consisted of two layers of muscle fibres (1,2) separated by a thin layer of collagen (black arrowhead) enclosing a wide lumen (lu). Scale bar = 30 μm. **e** TEM of a cross-section through the proximal venom gland. The cytoplasm of the venom gland cells was filled with small, spherical secretory granules (see Supplementary Fig. 2b, c). The dissociation of epithelial cells from the basal lamina (white arrowhead) and rupture of their membrane indicate holocrine secretion of venom granules into the lumen (lu). Scale bars = 10 μm. bc buccal cavity, bt buccal tube, lu lumen, m mouth, n nucleus, oe oesophagus, pr proboscis, rsl long arm of radular sac, rss short arm of radular sac, vb venom bulb, vg venom gland. Four juveniles at 6 dps were sectioned, with one used for the 3D reconstruct and one used for TEM.

wild-caught specimens[23]. The hooked accessory process and the basal ligament seen in the adult tooth were also absent. The juvenile radular tooth was short in absolute and relative length, measuring $69.7 \pm 1.15$ μm ($n = 5$) in length for a shell length ($S_L$) of $1.71 \pm 0.08$ mm ($n = 5$) (4.1% of $S_L$). It had a waist and a broad base with a wide opening, as typically seen in vermivorous species. Interestingly, similar teeth are also found in juvenile worm-[27] and mollusc-hunters (Rogalski, A. et al., manuscript in preparation), indicating that this trait has been retained in early life stages across Conidae. Morphometric analyses confirmed similarity with radular teeth from vermivorous cone snails (Supplementary Fig. 3; Supplementary Data 1), and the presence of similar teeth in related conoidean lineages such as Mitromorphidae and Borsoniidae[28,29] suggests this trait may be plesiomorphic within the group.

In contrast, adult *C. magus* are strict piscivores and catch prey using a "hook-and-line" foraging behaviour[30] (Fig. 4c). This strategy relies on a harpoon-like radular tooth (Fig. 4d; Supplementary Fig. 4a) to deliver potent neurotoxins that target the skelektal musculature of the fish, producing immediate rigid paralysis characterised by continuous extension of the fins[30,31]. The strong accessory process (recurved barb) at the apex of the tooth allows to tether the fish securely, which is then withdrawn into the rostrum and swallowed whole. The adult radular tooth was long ($3.4 \pm 0.1$ mm; $n = 5$) ($S_L = 53$ mm) (6.4% of $S_L$), with a small base attached to a flexible ligament. This trait is thought to have evolved convergently in Atlantic/Eastern-Pacific and Indo-Pacific fish-hunting cones that employ a similar "hook-and-line" foraging behaviour[30,32], suggesting the strong adaptive value of this character.

## Transcriptomics reveals venom ontogeny

To compare the expression of conotoxin genes during ontogeny, de novo transcriptomes were generated from embryonic ($n \sim 500$), juvenile ($n = 2$) and adult ($n = 1$) *C. magus*. After quality filtering, 5,410,000 reads obtained from the sequencing of two juvenile *C. magus* were assembled into 140,457 contigs of an average length of 468.3 nucleotides. Transcriptome annotation using blastx and blastp searches resulted in the identification of 68 full-length toxin-encoding transcripts (59 mature peptides, 10 cysteine frameworks). Among these, 58 could be classified into 16 conotoxin gene families, dominated by the O1, H, M, and O2 superfamilies (47.1% of transcripts, 60.4% of combined expression) (Fig. 5a; Supplementary Data 2). Juvenile *C. magus* also expressed a number of hormone-like conopeptides ($n = 10$) that combined contributed 1.7% of total expression. These included two consomatins (consomatin_M2 and M3), a family of somatostatin-mimicking peptides that have been recruited into the venom arsenal of many cone snail species, likely to facilitate prey capture[33]. Consomatin_M2 and consomatin_M3 shared similarity with annelid signalling somatostatin-related peptides (Supplementary Fig. 5), suggesting these might be used to hunt worms in juveniles before being downregulated in adulthood, as previously hypothesised[33]. Consistent with their diet, highly expressed juvenile transcripts shared high sequence similarity with vermivorous conotoxins (Supplementary Fig. 6), indicating a strong link between feeding ecology and venom biochemistry. Since heterogenous toxin distribution along the VG has been widely documented in Conidae[34–36], the adult VG was divided into proximal and distal regions which were sequenced independently, yielding 5,030,000 and 5,590,000 high-quality reads that were assembled into 87,157 and 97,528 contigs of an average length of 487.8 nucleotides. Annotation of the maternal VG transcripomes led to the identification of 69 unique full-length toxin-encoding transcripts (52 mature peptides, 9 cysteine frameworks). Among these, 61 could be classified into 12 conotoxin gene families, with the proximal VG dominated by the O1, M and T superfamilies (58.1% of transcripts, 63.7% of expression), while the distal VG was

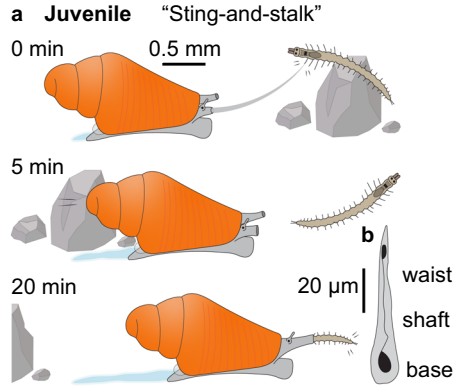

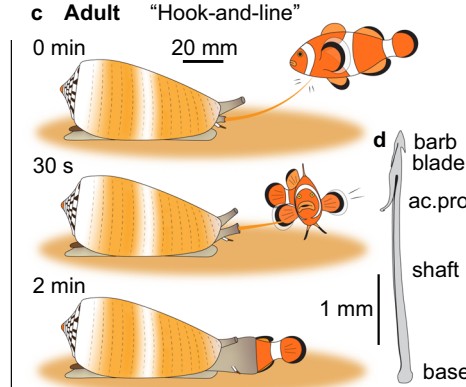

**Fig. 4 | Feeding behaviour of juvenile and adult *C. magus*.** The transition to adulthood in *C. magus* was marked by a shift in prey preference accompanied by morphological and behavioural changes. **a** Early juveniles were observed to feed exclusively on syllid worms in a "sting-and-stalk" foraging behaviour. Worms were first stabbed by a short unbarbed radular tooth, and the proboscis rapidly withdrawn inside the rostrum, leaving the prey untethered. Venom injection induced hypoactivity in worm prey. The juvenile then stalked its prey before swallowing it whole. **b** Juvenile radular tooth redrawn from scanning electron microscopy (SEM) (see Supplementary Fig. 2a). This "sting-and-stalk" foraging behaviour was consistent with the juvenile radular tooth lacking apical barbs, blades and serrations.

The juvenile radular tooth was short in absolute and relative length and was characterised by a short shaft, a waist and a broad base with a wide opening. **c** Adult *C. magus* display the typical "hook-and-line" foraging behaviour, with the fish prey being stung by a hooked radular tooth while potent neurotoxins rapidly induce rigid paralysis. The paralysed fish is then tethered inside the rostrum and swallowed whole. **d** Adult radular tooth redrawn from SEM (see Supplementary Fig. 4a). Note the presence of the hooked accessory process (ac.pro) at the apex of the tooth that facilitates tethering stung fish. Radular teeth are magnified 25x relative to the size of the animals.

dominated by the A, O1 and M superfamilies (61% of transcripts, 83.5% of expression) (Fig. 5a; Supplementary Data 2). These findings were consistent with recent studies where the M, O1, T and A superfamilies dominated the VG transcriptomes of adult *C. magus* specimens from Japan and the Philippines[37], suggesting different populations of this species share a similar repertoire of conotoxins in adulthood. The adult VG also comprised a number of hormone-like conopeptides (*n* = 8) which combined contributed 1.2% and 14.1% of expression in the proximal and distal regions, respectively. Remarkably, only 10 conotoxin precursors were shared between the juvenile and the maternal adult VG transcriptomes, with five precursors shared between our juvenile transcriptome and adult specimens from Japan and the Philippines[37]. Principal component analysis (PCA) confirmed the differential expression of conotoxin gene families between juvenile (*n* = 2) and adult (*n* = 2) *C. magus* (Fig. 5b). The lowest PCs (PC1 and PC2) accounted for 52.9% and 31.8% of this variation, driven mostly by overexpression of the A, M and T superfamilies in adults and overexpression of the O1, L and B2 superfamilies in juveniles (Supplementary Data 3).

While the O1 and M superfamilies accounted for a large proportion of transcripts in both juvenile and adult *C. magus*, conotoxins belonging to these superfamilies showed remarkable diversity in their signal sequences and encoded mature peptides. Based on signal sequence similarity, M conotoxins can be divided into two groups[38]. All embryonic and juvenile M conotoxins (*n* = 9) belonged to the MLKM group, whereas adult *C. magus* also included precursors from the MMSK group (Supplementary Data 2). The mature sequences of M conotoxins (III cysteine framework, CC-C-C-CC) can be further divided into five branches (M-1 to M-5) based on the number of residues between the fourth and fifth cysteines[39]. Both proximal and distal adult VG expressed a precursor from the M-4 branch (M_M3.6i), which is found in piscivorous cone snails and includes κM-, ψ- and μ-conotoxins that induce paralysis in fish[39]. This precursor, along with two other precursors encoding the same mature peptide (PMAG100 and PMAG106), were also found in adult *C. magus* from Japan and the Philippines[37]. In contrast, the juvenile transcriptome included only M-1 branch conotoxins that are primarily found in mollusc- and worm-hunting cone snails and known to elicit a spasmodic response in prey[38,40] (Supplementary Data 2).

To explore the divergence of O1 conotoxins between juvenile and adult *C. magus*, we performed a phylogenetic analysis. (Supplementary Fig. 7). While all O1-superfamily precursors shared the same VI/VII cysteine framework (C-C-CC-C-C), our phylogenetic history was consistent with previous studies that split the superfamily into four paralog groups with distinct pharmacologies[37]. Notably, new and highly expressed juvenile O1_M6.53 and O1_M6.54 (49.7% of combined expression) bifurcated early within the O1-2 paralog group, which comprised vermivorous conotoxins of unknown function, while the presence of lowly expressed adult precursors (<1.5% of expression) within this group suggests these may be downregulated in adulthood. Supporting the divergence of juvenile conotoxins, their conserved signal and prepropeptide regions differed from the corresponding adult precursors from the same paralog group. In contrast, highly expressed adult precursors clustered within the O1-1 (35.4% and 14.4% in proximal and distal VG, respectively) and O1-3 (0.5% and 18.6%, respectively) paralog groups that include ω- and δ-conotoxins targeting vertebrate calcium and sodium channels, respectively[41,42] (Supplementary Fig. 7). The diversity of conotoxins from piscivorous species within these paralog groups supports the idea that gene duplication events and subsequent functional divergence within this gene family facilitated the adaptive evolution of piscivory[43,44]. We identified a putative δ-conotoxin (O1_M6.61ii; VI/VII cysteine framework) clustering within the O1-3 paralog group that was shared between embryonic, juvenile and adult transcriptomes. This variant of δ-MVIC (91% similarity in the mature peptide region) shared high sequence similarity with δ-conotoxins from other piscivorous cone snails, including a conserved hydrophobic patch important for activity on vertebrate sodium channels[45].

Venom ontogeny in *C. magus* was also marked by the diversification of the A superfamily. As typically seen in piscivorous species[46–49], this superfamily was structurally and functionally diverse in adult *C. magus*, including highly expressed κA-conotoxins (IV cysteine framework, CC-C-C-C-C) that cause hyperexcitability of axons at the venom injection site, resulting in rapid rigid paralysis of the prey[9]. High expression values of κA-conotoxins in the adult VG (25.5% and 47.6% in proximal and distal VG, respectively) was consistent with the symptoms observed in fish prey during feeding (Fig. 4c; Supplementary Data 2). We identified two new putative κA-conotoxins that

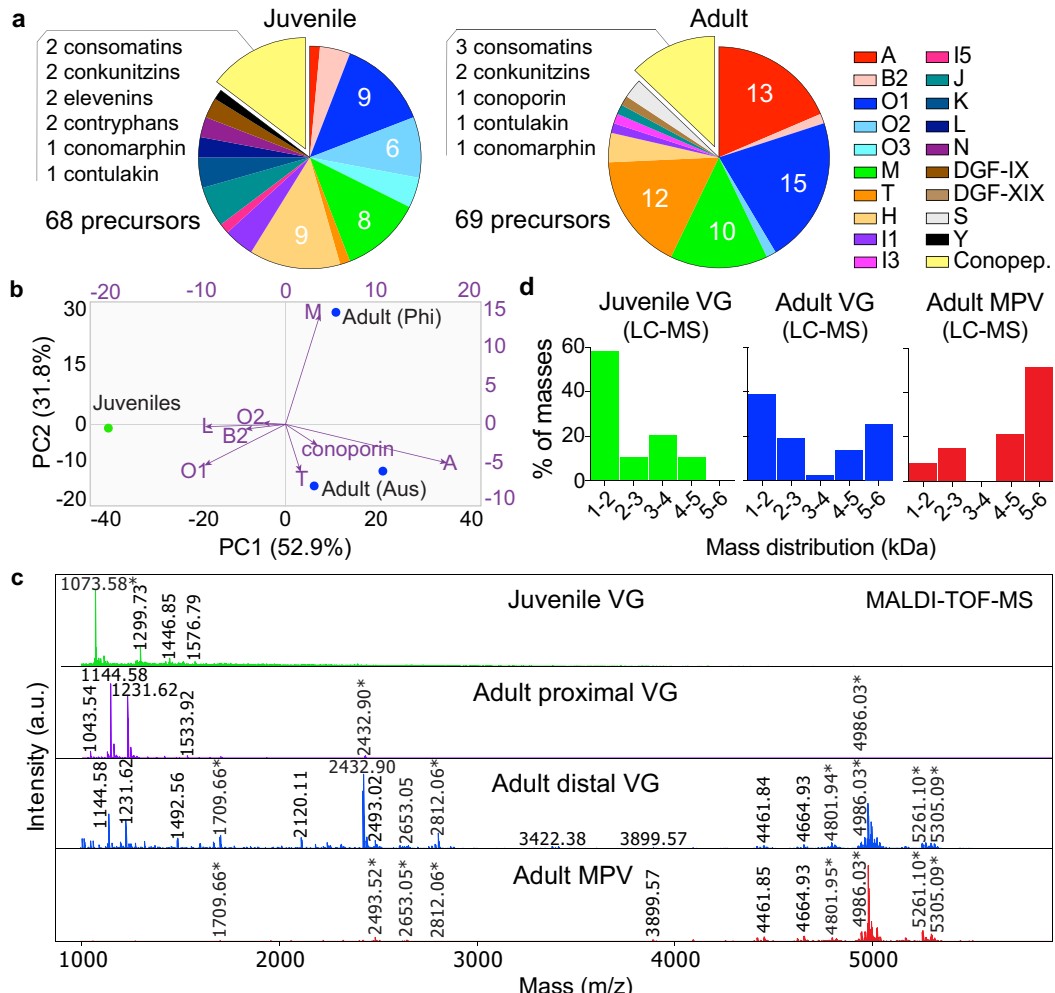

**Fig. 5 | Venomics reveals venom ontogeny in *C. magus*.** Combined transcriptomics and proteomics revealed venom ontogeny in *C. magus*. **a** Distribution of conotoxin and hormone-like conopeptide (Conopep.) precursors in the juvenile and adult transcriptomes. The juvenile venom transcriptome was dominated by the O1, H, M and O2 superfamilies, whereas the adult venom transcriptome was dominated by the O1, A, T and M superfamilies. **b** Principal component analysis biplot comparing venom composition between juvenile (green) and adult (blue) *C. magus* from Australia (Aus) and the Philippines (Phi). Arrow length indicates how strongly each variable (conotoxin gene superfamilies) influences the principal components (PC). Left and bottom axes: PC scores of samples. Right and top axes: loadings of variables. The position of juvenile *C. magus* at the extreme of PC1 is driven by the overexpression of the O1, L and B2 superfamilies, while the placement of adult specimens is largely explained by the overexpression of the A, M and T superfamilies. **c** High-resolution MALDI-TOF-MS revealed different peptide compositions between the juvenile and adult VG, and the heterogenous distribution of conotoxins along the adult VG. Asterisks indicate masses also found in LC-MS experiments in the corresponding samples. **d** Distribution of monoisotopic masses (≥0.1% of expression) in the juvenile and adult VG proteomes. Source data for transcriptomics, PCA and proteomics are provided as Supplementary Data 2–4, respectively.

contained conserved threonines at positions 7, 9 and 10 that are often post-translationally glycosylated[50]. A_M4.7ii, a variant of κ-MIVA (94% similarity in the mature peptide region), was found in juvenile and adult *C. magus*, while A_M4.6 was unique to embryos. The A superfamily also included inhibitory α-conotoxins (I cysteine framework, CC-C-C) that target nicotinic acetylcholine receptors to induce persistent flaccid paralysis in vertebrates[51,52]. These could be further assigned to the α3/5 (A_M1.17), α4/7 (A_M1.2) and α4/4 (A_M1.7) subgroups. In contrast to κA-conotoxins, α-conotoxins were restricted to the adult VG (Supplementary Data 2).

Despite feeding on worms, high sequence similarity with characterised conotoxins and conserved cysteine frameworks suggests that juvenile κA- and δ-conotoxins may act on vertebrate ion channels. Indeed, vertebrate-active δ-conotoxins previously identified in adult vermivorous cone snails were hypothesised to have first arisen for defence and later repurposed to facilitate the evolutionary shift from worm- to fish-hunting[53–55]. However, whether juvenile κA- and δ-conotoxins may be used to deter vertebrate predators and/or facilitate predation on worms remains to be determined. We additionally identified a novel juvenile O2 conotoxin (O2_M6.62; VI/VII cysteine framework) that shared high sequence similarity with γ-conotoxins from molluscivorous species but lacked the −ECCS− motif where the glutamic acid is post-translationally modified to a γ-carboxyglutamate[42] (Supplementary Fig. 6). This suggests that γ-conotoxins producing strong paralytic effects in molluscs[42] may have first evolved to deter molluscan predators before being repurposed for predation in molluscivorous lineages.

Additionally, 18 unique conotoxins displaying the VI/VII cysteine framework and with undefined pharmacology were retrieved from the juvenile transcriptome (Supplementary Data 2). This cysteine arrangement is typically associated with the inhibitor cysteine knot (ICK) structural motif[56]. Peptides adopting this fold display a range of biological activities, including the clinically used N-type calcium channel inhibitor ω-conotoxin MVIIA (Prialt®)[22,57], highlighting the potential of juvenile cone snails as an untapped source of novel bioactive peptides.

## Juvenile and adult *C. magus* secrete distinct VG proteomes

To further investigate venom ontogeny in *C. magus*, the juvenile ($n = 20$) and adult ($n = 1$) VG proteomes were compared by mass spectrometry. Matrix-assisted laser desorption/ionisation time-of-flight mass spectrometry (MALDI-TOF-MS) revealed that juvenile and adult VG proteomes were dominated by distinct suites of peptides <6 kDa, with masses >4 kDa restricted to the adult VG (Fig. 5c; Supplementary Fig. 8a; Supplementary Data 4). Furthermore, the different MS patterns obtained from proximal and distal VG support the heterogeneous distribution of conotoxins along the adult VG. While MALDI-MS is a useful technique for whole venom profiling, this approach suffers a number of limitations, including low dynamic range and ion suppression effects, preventing the detection of the full venom complexity[58]. To complement MALDI-MS, we additionally performed liquid chromatography-mass spectrometry (LC-MS) on the juvenile and adult *C. magus* VG extracts. Considering the complexity of cone snail venoms and the typical mass range of conotoxins, only monoisotopic masses between 1–10 kDa and covering ≥0.1% of relative intensity were considered to facilitate ecological interpretation (Supplementary Data 4). A total of 123 masses (104 unique) were detected in the adult VG, while 92 masses (86 unique) were found in the juvenile VG. Comparison of mass lists revealed only a single mass (1438.01 Da) was shared between both venom proteomes, supporting the differences observed by MALDI-MS. While the juvenile VG proteome was largely dominated by peptides falling into the 1–2 kDa mass range ($n = 53$, 57.6% of masses), the adult VG proteome contained a large proportion of 4–6 kDa peptides ($n = 48$, 39% of masses) compared to juveniles ($n = 10$, 10.9% of masses) (Fig. 4d; Supplementary Fig. 8b).

To identify conotoxins used for the capture of vertebrate prey by adult *C. magus*, the milked predatory venom (MPV) was obtained from the laying female using a fish to elicit a predatory response[59]. Despite the diversity of venom peptides found within the maternal VG, the MPV was dominated by peptides in the 4–6 kDa mass range (76.5% of masses) eluting at 25–30% acetonitrile (Fig. 4e; Supplementary Figs. 8, 9). This mass range and elution window are typical of excitatory O-glycosylated κA-conotoxins, which dominate the predatory venom of sister species *C. catus* and *C. striatus*[48,49]. Glycan groups identified at 204.08 (HexNAc), 366.13 (HexHexNAc) and 528.19 (Hex2HexNAc) *m/z* were prominent in this elution window in both the adult VG and MPV (Supplementary Fig. 9). These findings indicate that the MPV of adult *C. magus* is also dominated by glycosylated peptides, presumably κA-conotoxins, which was consistent with their high transcriptomic expression. In contrast, the MPV lacked the more hydrophobic venom components and the smaller molecular weight peptides present in either region of the adult VG, as previously observed in the piscivorous *C. consors*[60] (Fig. 5c; Supplementary Fig. 8).

The presence of κA-conotoxins in adult *C. magus* was further supported by liquid chromatography-tandem mass spectrometry (LC-MS/MS), with the identification of κA-like conotoxins A_M4.5 in the MPV and A_M4.7 in the adult VG (Supplementary Table 1), the latter consistent with its high transcriptomic expression in both the proximal (4.3%) and distal (25.5%) VG. Other peptides identified in both the adult VG and MPV included O1 conotoxins (O1_M6.15, O1_M6.60, O1_M7.4 and ω-conotoxins MVIIA [Prialt®] and MVIIB), S conotoxins (S_M8.1 and S_M8.6), M conotoxin M_M3.4 (MMSK group), T conotoxin T_M6, conkunitzins (conkunitzin_M9.10, conkunitzin_M9.12, conkunitzin_M14.9) and conoporins (conoporin_M1 and conoporin_M6). Consistent with our RNA-seq experiments, α-conotoxin MII (A_M1.2), α-like conotoxin A_M1.17 and δ-like conotoxin O1_M6.61 were also found within the adult VG but were missing from the MPV (Supplementary Table 1). Unfortunately, MS/MS on the juvenile VG was not feasible due to the limited sample available.

Our transcriptomis and proteomic analyses support the heterogenous distribution of conotoxins along the adult VG, consistent with early studies of *C. magus* showing the proximal VG extracts induced flaccid paralysis in fish and mice, while distal VG extracts induced rigid paralysis[30]. Supporting this spatial expression pattern, the secretory cells lining proximal and distal regions of the adult VG differed in their ultrastructure and secretory products (Supplementary Fig. 4b, c). In contrast, the juvenile VG appeared structurally homogenous (Supplementary Fig. 2b, c), but functional characterisation of proximal and distal regions was impeded by the small size of the juvenile VG (<0.5 mm). The extent to which the structural compartmentalisation of the developing VG coincides with the diversification of vertebrate-active conotoxins and the shift from vermivory to piscivory remains to be fully elucidated.

In conclusion, this study has revealed that the shift from worm- to fish-hunting during the development of *C. magus* is underpinned by complex morphological, behavioural and molecular changes. This parallel between ontogeny and phylogeny echoes Haeckel's biogenetic law[61] and our findings support previous evidence that fish-hunters evolved from vermivorous ancestors[11,12]. Combined transcriptomic and proteomic approaches revealed the expression of prey-specific conotoxins across the life history, indicating a strong link between feeding ecology and venom composition, as seen in other venomous taxa[62–64]. The distinct composition of juvenile venoms provides new insights into the evolution of conotoxins and new opportunities for the discovery of venom peptides with novel structures and functions, even in extensively-studied species. Our investigation of captive-bred specimens provides new opportunities for comparative genetic studies and could help reduce reliance on wild populations, which are intensively exploited for the ornamental shell trade[65].

## Methods

### Source and culture of larvae and juveniles

Adult specimens of *Conus magus* from the Great Barrier Reef (Queensland, Australia) were sourced from Cairns Marine (Cairns, Queensland, Australia) and maintained in 8 L breeding tanks in a PC2 laboratory on a closed saltwater recirculating system. Water temperature was set as $26 \pm 1\,°C$, with 12:12 hours light-dark (LD) cycle. Water parameters and animal health were monitored daily and food (zebrafish) was provided on a weekly basis. Use of zebrafish (*Danio rerio*) in this study was approved by the University of Queensland Animal Ethics Committee (approval number 2019/AE000271). Coral rocks (Cairns Marine) were added into the tanks to provide a substrate for egg attachment. Egg capsules were removed from the tank on the day following oviposition, rinsed in ultra-filtered (0.22 μm) sea water and placed in UV-sterilised 1 L tanks filled with ultra-filtered sea water sourced from Caloundra, Queensland. The culture tanks were maintained under a 12:12 hours LD cycle, at room temperature ($24 \pm 1\,°C$). Each day, a third of the water was replaced with fresh ultra-filtered sea water until hatching. Penicillin/Streptomycin antibiotics (50 μg/ml) were added weekly to the culture tanks to minimise bacterial contamination. Within 24 h of hatching, larvae were gently pipetted into UV-sterilised 1 L tanks, filled with 10 μm-filtered sea water at a concentration of 25 larvae/L. They were fed daily a mix made of equal amounts of live *Tisochrysis lutea* and *Chaetoceros muelleri* (Australian National Algae Culture Collection, CSIRO, Australia) at a concentration of $15 \times 10^3$ cells/ml. Larvae were transferred to clean culture tanks every second day by manual pipetting, with dead and unhealthy specimens identified visually and discarded. Larvae were cultured under a 12:12 hours LD cycle, at room temperature ($24 \pm 1\,°C$) without added antibiotics. Late larvae were induced to settle on CCA-covered rocks sourced from the Great Barrier Reef (Cairns Marine). Early juveniles were transferred to UV-sterilised 1 L tanks of ultra-filtered sea water (50% replaced daily) and CCA-covered rocks added as a source of food. Images of live specimens (larvae and juveniles) were captured on a Leica EZ4 stereomicroscope (Leica Microsystems) using LAS EZ 3.0.0 software (Leica Microsystems), with image editing (brightness and contrast adjustments, background removal) performed on Photoshop 21.2.1.

## Culture of microalgae

*Tisochrysis lutea* and *Chaetoceros muelleri* were maintained aerobically in 500 ml sterile flasks in artificial sea water enriched with f/2 culture medium[66]. Culture flasks were kept inside an incubator at 24 °C under a 14:10 hours LD cycle. Light was provided by fluorescent tubes at an intensity of 80 µE.m$^{-2}$.s$^{-1}$. Primary stock cultures were sub-cultured every week.

## Histology and computer-assisted reconstruction

Prior to fixation, specimens for histology were anaesthetised by incubation in high Mg$^{2+}$/low Ca$^{2+}$ artificial sea water[67] for 60–90 min before being placed on ice for 5–10 min. Subsequent fixation, decalcification, dehydration and resin infiltration were conducted with vacuum microwave assistance (Pelco). Specimens were fixed and stored in 2.5% glutaraldehyde in sea water at pH 7.5 for up to 2 weeks before processing. Larval shells were decalcified in a 0.5% formic acid (FA) solution, whereas the thicker shells of metamorphic stages and juveniles were carefully removed with the aid of forceps. Decalcified specimens were rinsed in Phosphate Buffer Saline (PBS) before post-fixation in cacodylate-buffered 1% osmium tetroxide. Specimens were subsequently dehydrated in an ethanol dilution series and infiltrated and embedded in Epon resin (ProSciTech). Serial cross-sections were obtained for 1, 7 and 15 dph larvae, and 1, 2 and 6 dps juveniles (2–4 specimens at each stage). For light microscopy, sections were cut on an ultramicrotome at a thickness of 1 µm using a Diatome histo-knife, stained with toluidine blue, and imaged with a DP70 CCD camera mounted on a BX51 upright widefield microscope (both Olympus) using Zen Blue 3.2 software (Zeiss). For transmission electron microscopy, sections were cut at a thickness of 80 nm, collected on copper grids, and imaged on a Hitachi HT 7700 fitted with a XR401 high sensitivity CMOS camera (AMT) using TEM System Model HT7700 02.30.15.56 software (Hitachi).

The three-dimensional (3D) reconstruction of the juvenile venom apparatus was obtained from serial cross-sections imaged on an AxioScan Z1 slide scanner using Zen 3.5 software (both Zeiss). Individual images were cropped and stacked using ImageJ/Fiji 2.0.0 with further image analysis, visualisation and reconstruction performed using Amira 2021.1 before sections were aligned and annotated manually. The 3D volume was then generated and smoothed using Amira and Photoshop 21.2.1.

## Scanning electron microscopy of radular teeth and data analysis

Mature radular teeth were isolated from the radular sac of dissected specimens and briefly washed in a 10% sodium hypochlorite solution. The teeth were subsequently rinsed in two washes of UHQ water. Adult radular teeth were dehydrated in an ethanol dilution series before immersion in two changes of hexamethyldisilazane (HMDS). Following the second HMDS wash, adult teeth were air-dried at room temperature for 30 min, while juvenile radular teeth were directly air-dried following the UHQ washes. Radular teeth were then mounted on aluminium pin stubs using carbon adhesive discs and coated with platinum in a CCU-010 sputter coater (Safematic). Images were captured with a TM4000Plus SEM using TM4000 Tabletop Microscope 1.5 software (both Hitachi). Morphometric data for radular teeth were plotted using Prism 9.4.1 (GraphPad). Statistical significance was defined as $P < 0.05$ using a one-way ANOVA test and data are presented as means ± SEM.

## RNA extraction and sequencing

For RNA-sequencing, total RNA was extracted from whole-body embryos (7 days post-oviposition, $n \sim 500$), juveniles (14 dps, $n = 2$, $S_L = 1.7$ mm) and the maternal VG ($n = 1$). Embryos and juveniles were placed into > 10-fold the tissue volume of RNA later (Invitrogen) and stored at −80 °C until extraction. The maternal VG was dissected and divided into proximal- and distal-regions of equal sizes to investigate

spatial distribution of conotoxins along the VG and RNA extracted from fresh tissue. Three segments corresponding to proximal, central and distal regions were kept in a solution of 30% acetonitrile (ACN)/1% FA for proteomics, and two small segments (proximal and distal) were placed in 2.5% glutaraldehyde and processed for histology as described above. Total RNA was extracted from all samples using TRIzol (Invitrogen) following the manufacturer's instructions to yield 0.4–2.72 µg of purified mRNA from each sample. The RNA quality and concentration were assessed on a 2100 Bioanalyzer using the RNA 6000 Nano kit (Agilent). Complementary DNA library preparation and sequencing were performed by the Institute for Molecular Bioscience Sequencing Facility (University of Queensland). Libraries were constructed using the Illumina Stranded mRNA Prep kit. Samples were pooled in a batch of 6 and 600-cycle (2 × 300 bp) paired-end sequencing was performed on an Illumina MiSeq instrument. Raw sequencing data have been deposited in the NCBI Sequence Read Archive under BioProject accession number PRJNA943605.

## Transcriptomic analysis

Raw data processing and de novo assembly were performed by the Queensland Facility for Advanced Bioinformatics (University of Queensland). Adapter trimming, quality trimming and filtering were performed using BBDuk (BBTools 38.84 package, Joint Genome Institute). Merging of paired-end reads and error-correction were then performed using BBMerge (BBTools). Trimmed and error-corrected reads were de novo assembled using Trinity 2.8.4[68] using *k*-mer sizes of −19 and −31. The assembled contigs from both *k*-mers were merged into single datasets and duplicates removed using Dedupe (BBTools). Assembled contigs were then searched against ConoSorter 1.1, an in-house program developed to classify conotoxins into gene superfamilies and classes[69]. After translating nucleic acid sequences into all possible reading frames, the algorithm isolates conotoxin coding sequences and classifies them using regular expression and profile hidden Markov models searches. The programme searches the ConoServer database[70] and provides additional information for matching sequences, including relative sequence frequency, length, number of cysteines, N-terminal hydrophobicity and sequence similarity score. Sequences containing <50 and >250 amino acids and with a signal region hydrophobicity score <45% were manually removed. All sequences were searched for the presence of an N-terminal signal region using the SignalP 5.0[71] server and sequences lacking signal regions were discarded. At this stage, nucleotide sequences were manually inspected and incomplete or aberrant sequences (internal or no stop codons, repetitions, incorrect open reading frames) were discarded. The retained contigs were annotated using blastx and blastp[72] searches against the non-redundant UniprotKB/SwissProt (*E*-value cut off: 10$^{-3}$) and Conoserver databases. The ConoPrec tool in Conoserver was then used to identify the signal-, propeptide-, mature- and post-mature regions and cysteine frameworks. Expression levels of all reads were computed in transcripts per million (TPM)[73] using Kallisto 0.46.1[74]. Expression levels were summed up for each gene superfamily and relative expression (in per cent) calculated, including a specimen from the Philippines[37]. We then performed a principal component analysis (PCA) to evaluate the degree of venom composition similarities between juvenile and adult *C. magus* using XLSTAT statistical software (Addinsoft, free trial version). For the PCA biplot, the four variables with the strongest influence on the PCs are shown. The data matrix, summary statistics, contribution of each variable (in per cent), PCA scores and loading plots can be seen in Supplementary Data 3. All peptide precursors were named according to the conventional conotoxin nomenclature (with species represented by one or two letters, cysteine framework by an Arabic numeral and, following a decimal, order of discovery by a second numeral)[75], with slight modification[76]. The superfamily was added as a prefix and precursors differing in their propeptide regions but with conserved mature

peptides were differentiated with a small roman numeral as a suffix to distinguish between minor variants. All conotoxin precursor sequences have been deposited in NCBI GenBank [https://www.ncbi.nlm.nih.gov/nuccore] under accession numbers OQ644315–OQ644445.

## Proteomic analyses

MS-based proteomic studies were carried out on the pooled VG of 20 juveniles at 14 dps and on the maternal VG (proximal, central and distal). All samples were extracted in 30% ACN/1% FA and kept at −20 °C until further analysis. Predation-evoked venom samples were obtained from the egg-laying female, as previously described[59]. Following each milking, the collecting tube was briefly centrifuged and stored at −20 °C until analysis. For our analysis, venom samples from 10 independent milking events and collected over 18 months were pooled.

In preparation for LC-MS/MS, each sample was reduced with triethylphosphine and alkylated with 2-iodoethanol[77], before digestion with proteomics-grade trypsin (Sigma). This step was omitted for the juvenile venom gland extract due to limited sample available. Native and digested samples were desalted using a $C_{18}$ ZipTip (Merk Millipore) and dried by vacuum centrifugation. The samples were reconstituted in 0.1% FA and 50–200 ng (estimated from $A_{280}$) were loaded on a nanoEase M/Z HSS T3 (Waters; 100 Å pore size, 1.8 μm particle size, 300 μm x 150 mm) analytical column and separated on a linear gradient of 3–60% solvent B (0.1% FA in ACN) in solvent A (0.1% FA) over 20.5 min at a flow rate of 5 μl/min. The LC outflow was coupled to a ZenoTOF 7600 mass spectrometer (SCIEX) equipped with an OptiFlow Turbo V ion source. The samples were analysed using the data dependent acquisition (DDA) method with data acquired in the positive-ion mode. MS scans were acquired across 400–1750 mass/charge ratio ($m/z$) over 200 ms, and data acquisition was performed on up to 20 candidate ions with a charge of +2 to +6 and signal > 150 counts/s. The most intense isotopes were selected and fragmented with collision-induced dissociation (CID) and electron-activated dissociation (EAD) tandem mass spectrometry. MS/MS scans were collected between 50–2000 $m/z$ over 35 ms. The dynamic collision energy setting was used, allowing collision energy to vary based on $m/z$ and $z$ of the precursor ion. Data were acquired using OS 3.0.0.3339 and analysed in Peakview 2.2 (both SCIEX). The CID-MS/MS spectra were searched against a database combining all translated sequences from our RNA-seq experiments and previously reported *C. magus* conotoxins (Supplementary Data 2) using the Paragon[78] algorithm implemented in ProteinPilot 5.0 (SCIEX) with the following settings: iodoethanol (for reduced and alkylated samples), trypsin digested (for digested samples), common conotoxin post-translational modifications[79], biological modifications, thorough ID. Peptides with ≥2 tryptic fragments at a confidence of 99 and a false discovery rate <1% were considered genuine. The EAD-MS/MS data were searched against the same database using Mascot 2.5.1[80] (Matrix Science) with the following settings: trypsin, 1 missed cleavage, carbamidomethyl as a fixed modification, oxidation of methionine and deamidation of asparagine and glutamine as variable modifications, 20 ppm peptide tolerance, 0.1 Da MS/MS tolerance, 2 + 3+ and 4+ peptide charges, with an error tolerant search included. Peptides with ≥2 tryptic fragments, individual peptide scores >60 and a significance threshold <0.05 were selected.

For MALDI-TOF-MS, 1 μL of each sample was spotted together with 2 μL of diluted α-cyano-4-hydroxycinnamic acid (CHCA) solution (working solution stored as an acetone-saturated solution and diluted 1 in 10 with a solution of ethanol/acetone/water [6:3:1] and 0.1% TFA) onto a polished steel target. Samples were analysed on a timsTOF fleX MALDI-2 (Bruker) operated in TIMS off (or QqTOF only) mode, controlled using TIMS Control 3.1.13.0. Positive mode analyses were undertaken over $m/z$ 1000 to 10,000, 10000 shots with laser repetition rate of 10,000 Hz. Calibration was performed using a single spot of CHCA with Bruker peptide calibration mixture and recombinant human insulin. Data were acquired using timsControl 3.1.4 and visualised using Data Analysis 6.0 (both Bruker). Heatmaps in Supplementary Figure 8 were built using Prism 9.4.1 (GraphPad).

## Phylogenetic analysis

Amino acid sequences of full-length precursors were aligned using the local paired iterative alignment method (L-INS-i) in MAFFT 7.504[81], and the quality of the alignments checked manually. The evolutionary history of the O1 superfamily was reconstructed using a molecular phylogenetic approach. The most appropriate evolutionary model was determined in PhyML 3.0 using an Akaike Information Criterion test[82]. IQ-TREE 2.2 was then used to reconstruct molecular phylogenies by maximum likelihood, and branch support values estimated by ultrafast bootstrap using 10,000 replicates[83,84]. Because taxonomic outgroups could not be designated we used midpoint rooting to root the trees.

## Reporting summary

Further information on research design is available in the Nature Portfolio Reporting Summary linked to this article.

## Data availability

Raw sequencing reads have been deposited in the NCBI Sequence Read Archive (SRA) with BioProject accession number PRJNA943605. Conotoxin precursor sequences have been deposited in NCBI GenBank [https://www.ncbi.nlm.nih.gov/nuccore] with accession numbers OQ644315–OQ644445. Mass spectrometry proteomics data have been deposited in the ProteomeXchange Consortium via the PRIDE partner repository with the dataset identifier PXD042133. Raw data for radular tooth morphometry, principal component analysis and proteomics are available as Supplementary Data 1, 3 and 4, respectively.

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

## Acknowledgements

The authors wish to thank Richard Webb (Centre for Microscopy and Microanalysis, UQ) for skilful assistance with the histology and electron microscopy; Alun Jones (IMB Proteomic Facility, UQ) and Amanda Nouwens (School of Chemistry and Molecular Biosciences, UQ) for technical assistance with LC-MS; Brett Hamilton (Centre for Microscopy and Microanalysis, UQ) for help with MALDI-MS; Ian Ross (IMB, UQ) for expertise relating to algal culture and for sharing equipment and facilities, the School of Biomedical Sciences Core Imaging Facilities (UQ) for providing training on Amira 2021.1 software and the University of Queensland's Biological Ressources for zebrafish and cone snail husbandry. This work was supported by an Australian Research Council (ARC) Discovery grant (DP200103087) (R.J.L.).

## Author contributions

A.R. & R.J.L. conceived the project and designed experiments. A.R. carried out animal and algal cultures, histology, microscopy, computer-assisted 3D reconstruction, transcriptomic, proteomic and phylogenetic analyses and wrote the first draft of the manuscript. S.W.A.H. contributed to the transcriptomic and proteomic analyses and edited the manuscript. R.J.L. provided funding and research facilities and wrote the manuscript.

## Competing interests

The authors declare no competing interests.
