## [Peer Review File · Nature Communications]

REVIEWER COMMENTS

Reviewer #1 (Remarks to the Author):

The reviewed manuscript is a result of a well-designed and executed study, which combined multiple complementary approaches to document dietary shift in a piscivorous cone-snail species *Conus magus*. The origin of fish hunting in cone-snails has long been linked to the re-purposing of the defensive venom in a vermivorous ancestor, and the reviewed study yields an experimental proof for this hypothesis by convincingly demonstrating how the species' anatomy, radular morphology and venom composition undergo a concerted change at the transition from juvenile to adulthood. I did not find obvious flaws in the experimental design, lab procedures and data analysis. The manuscript is concisely written, and its conclusions are well backed up by the data. The illustrations are clear, well prepared, and are sufficient for readers to follow. Therefore, in my opinion, the submitted study certainly is novel, and high quality contribution, its findings are important and well presented, and therefore the paper fulfils the publication criteria of Nature communications. Nevertheless, I have some questions/comments/suggestions mainly to the manuscript text, and I hope that by addressing these the authors would further improve the quality of their study.

In the intro, the authors write that '... broad interest contrasts with the scarcity of literature on early life stages'. This is a very general motivation, and it implies a very exploratory and descriptive study. However, the paper is actually quite focused on a particular aspect – coordinated alteration at multiple levels following transition from vermivory to piscivory (as follows from the title). It is an appealing evolutionary question, and I would rather make it a basis of the motivation statement, rather than amorphous 'filling a gap in knowledge'.

'The blind-end of the VG had differentiated into a conspicuous venom bulb (proximal end)' - I do not agree with this interpretation of proximal / distal. The term 'proximal' suggests proximity to something, and logically for a venom gland, which opens to the buccal mass, 'proximal' should denote proximity to the to the main axis of the digestive system (i.e. alimentary channel). But the blind-end of the VG (or consequently the muscular bulb) are situated at the most distanced from the buccal mass part of the venom gland, and should therefore be considered distal. This (mis)use of proximal / distal goes back to the paper of Dutertre et al. (2014), and I would expect that in the present study the authors prefer following same terminology to be consistent. Nevertheless, for me it is simply erroneous, especially taking into consideration that the authors undertake some quality development / morphological study, and so would use morphological terms carefully.

'Morphometric analyses confirmed homology with radular teeth from vermivorous cone snails (Supplementary Fig. 3), and the presence of similar teeth in early divergent lineages such as Mitromorphidae and Borsoniidae'. - The conoidean families Mitromorphidae and Borsoniidae are not more early branching than Conidae based on the latest molecular phylogeny of the superfamily (Abdelkrim et al. 2018). Therefore, I would rather refer to them as to 'related (to cone-snails / Conidae) lineages/families of Conoidea'.

The sentence 'If juvenile O1-conotoxins evolved for predation on errant polychaetes, they are likely to be conserved since their first recruitment in vermivorous ancestors.' – is worded as a hypothesis and the hypothesis that does not seem to hold, based on the long branches of the juvenile O1 conotoxins on the Fig. 6. Therefore, one would expect further discussion in the subsequent sentences. However, the following sentence moves on to the adult O1 conotoxins, and seem to lack logically link with the preceding one. So, some revision is needed to close the logic gap here.

The term 'predatory venom' at the page 10 needs some introduction/explanation.

The paragraph addressing the heterogeneous distribution of venom components across the vg does not seem to be well connected to the rest of the story. I think to better integrate this part, functional differences between the proximal and distal parts of the venom gland could be better placed in the ontogenetic context. If by obvious reasons differential expression of venom gene superfamilies across vg cannot be inferred for juveniles (as the gland is too small to further divide it), then some histological differences between the distal VS proximal juvenile vg can perhaps be mentioned to show the developmental basis of this heterogeneity. Furthermore, some brief explanation would be needed for the terms like 'excitatory', 'inhibitory', 'flaccid paralysis', 'rigid

paralysis', and how this information is relevant to the main story that authors tell. In the conclusion, the authors suggest that their results potentially provide 'a glimpse into the feeding strategy of cone snail ancestors, prior to the emergence of fish-hunting.' This would have been a valid point, if all living cone-snail species were piscivorous. But the majority are worm-hunters, and their adult ecology/behavior/transcriptomes/proteomes allow for a more accessible inference of the feeding strategy prior to the emergence of fish-hunting. Finally, there is a discussion context that authors ignored – comparison of the venom composition inferred from their individual with that in the previously profiled individuals of *Conus magus*. The obvious strength of the revised paper is the comparison among life history stages, that is enabled by i) controlled lab environment, and ii) closely comparable genetic background of the studied individuals resulting from the fact that they represent progeny of one female. Whereas this design allows focusing on the signal of the changes that take place during the ontogenesis, it clearly diminishes variability in the individual development. This isn't a flaw at all, as I said, but it is something that deserves being discussed. The set of questions to be addressed: 'How representative is the observed pattern for the studied species? Or across fish-hunting lineages of cone-snails?' While the present work is pioneering in breeding cone-snails in lab environment, currently there are no comparable published data sets available. There is, however, published high quality transcriptomic data on the adult venoms of *C. magus* specimens from other geographic locations (Japan, the Philippines) (Pardos-Blas et al., 2019), and I feel that by using this data for comparative purpose, authors can address variability / plasticity of venom composition among individuals / populations.

At the end I would like to congratulate authors with a great research results,

Sincerely,

Alexander Fedosov

Reviewer #2 (Remarks to the Author):

The manuscript by Rogalski et al. interrogates the life history of *Conus magus* from eggs to metamorphosis. The research findings presented here are challenging to obtain as they rely on the ability to maintain the snails until egg laying and culture the embryos throughout hatching and metamorphosis. I think it's a beautiful study that will be well-received in the field and attract attention from related fields of inquiry.

Not only does this study expand our understanding of ontogenetic changes of the venom composition of one of the most-studied cone snail species but provides an innovative approach for the identification of new toxins that can be directly tracked to prey taxa.

Having said this, I have a few major concerns, that I would like to see addressed, one of which is the proper acknowledgement of the previous finding that *Conus magus* juveniles feed on worm and adults feed on fish. In the current version, the previous work by Nybakken and Perron is only marginally mentioned and not even discussed.

Additionally, as outlined below, all RNA-Seq raw data generated in this study have to be submitted to public repositories and accession numbers should be provided during revision.

Data presented in Figure 5 needs more rigorous, statistical analysis otherwise it is difficult to validate whether the data truly supports the claims.

Finally, some of the conclusions or hypothesis that were formulated in the paper are not well-supported by the data presented here and should be toned down. I have provided more detailed comments below.

Main Concerns:

All RNA-Seq data (and possibly also MS data) needs to be made available prior to acceptance of the manuscript. This encompasses all raw Illumina sequencing data (to be submitted to the SRA or ENA) and potentially also raw mass spec data (to be submitted to Pride or a similar public mass spec data repository). I am surprised that the authors did not already do this prior to submission. Either in the abstract or following the paragraph ending in line 46 it is imperative to properly cite and acknowledge the previous study on the shift from worm-hunting to fish-hunting behavior in juvenile and adult specimens of *Conus magus* (Nybakken, J. & Perron, F. E. Ontogenetic change in

the radula of *Conus magus* (Gastropoda). *Mar. Biol.* 98, 239–242 (1988)). Currently, this work is not mentioned until line 204 where the main findings and hypothesis presented in that paper are not properly presented (“Predation on polychaetes was consistent with worm setae identified in the stomach content”). This prior work needs to be presented in the abstract or introduction and should also be covered in the discussion.

Figure 5: there’s no easy way to assess potential significance in the data presented in Figure 5 without more information on statistical analysis in the method section and in the figure caption. Additionally, currently there is no information and no data showing how the peptides in panel e were identified and what the scale is that was used for the heat map. The authors should provide MS/MS evidence for the identified peptides as mass-matching cannot unambiguously identify cone snail venom peptides. If this data is not available, information on the observed vs. the calculate monoisotopic mass should be provided and it should be explained how the authors know where on the HPLC run these peptides elute. Overall, the data presented in figure 5 should be more rigorously interrogated and presented.

In line with this I find the analyses presented under “Juvenile and adult *C. magus* secrete distinct VG proteomes” underwhelming. Mass matching and intact mass comparisons are really not ideal for sample comparison, especially when better methodologies (such as MS/MS sequencing) are available. Please shorten and tone down these findings or re-analyze the samples using more accurate MS-based sample comparison methods.

Minor concerns/comments:

Lines 31-32: Do all cone snails have a planktonic stage? I thought that some transition into a benthic stage without a prior planktonic stage.

Lines 32-34 Clarify that not all cone snails have harpoon-like radula teeth.

Line 9: Cite Louise Page 2012 after “the foregut...”.

Line 36: these are more accurate references for research on the shift in prey type:
<https://pubmed.ncbi.nlm.nih.gov/24878223/>
and <https://www.sciencedirect.com/science/article/abs/pii/S0024406601905449>

Line 56: fix typo (“un”) and clarify that these toxins could have also been retrieved by exon capture or genome sequencing (e.g., an untapped source of novel bioactive venom peptides that would otherwise only be accessible through exon capture or genome sequencing).

Line 62: clarify how the species and sex was identified here (many snails are hermaphrodites and this should be clarified here).

Figs 1-3 and Supporting Figs 1-2 are beautiful and very informative presentations of the *C. magus* life history. Wonderful work!

Lines 220-22: This is a very intriguing hypothesis, and I am looking forward to seeing the findings on juvenile mollusk hunters.

Transcriptome analysis: it is not clear from the results or methods section how many individual adult venom glands were sequenced. This is important to mention and discuss as there can be significant intraspecies variation in venom gene expression. I think the data presented here goes beyond what would be expected for intraspecies variation, but nevertheless, it’s important to address this issue with scientific rigor.

Lines 278-286: very nice finding on the differences in the M-superfamily toxins

Lines 320 - 322: I disagree that the low expression of alpha conotoxins in the predatory venom suggests that there are used in defense. There would be several alternative hypotheses, such as prey-specific use of alpha conotoxins or a lack of reproducing natural predatory behavior in the lab. Either remove this sentence or provide potential alternative explanations.

Lines 330-331: It's not clear to me what study is being referred to here. Provide reference or clarify whether there's any data showing that these toxins are actually injected into worms.

Line 359: Provide confidence values for alphafold structures.

Lines 411-414: The differences in the data presented for the distal vs the proximal VG are weak and any conclusions from this data should be removed. There may be larger differences in other species but, it does not seem that this is the case here.

Lines 433-434: Several previous studies have provided phylogenetic evidence supporting that ancestral Conus were worm hunters. Thus, I strongly suggest changing the text from "support the hypothesis..." to "support previous evidence...". These papers should be cited here:
<https://pubmed.ncbi.nlm.nih.gov/24878223/>
and <https://www.sciencedirect.com/science/article/abs/pii/S0024406601905449>

I love the supporting video of the juvenile (393262_0_video_6976165_rjkpj4.mp4)! Is it be possible to label

RESPONSE TO REVIEWERS' COMMENTS

We thank the Reviewers for their supportive and insightful comments that have helped us to significantly improve the manuscript. Our revisions include both editing of the manuscript and the inclusion of new experimental MS data.

A detailed response to each issue raised is given below (in blue text). In summary, we have been able to address all the concerns raised by Reviewers 1 and 2 and trust our carefully revised manuscript is now suitable for publication in *Nature Communications*.

ADDRESSING REVIEWER 1 COMMENTS:

The reviewed manuscript is a result of a well-designed and executed study, which combined multiple complementary approaches to document dietary shift in a piscivorous cone-snail species *Conus magus*. The origin of fish hunting in cone-snails has long been linked to the repurposing of the defensive venom in a vermivorous ancestor, and the reviewed study yields an experimental proof for this hypothesis by convincingly demonstrating how the species' anatomy, radular morphology and venom composition undergo a concerted change at the transition from juvenile to adulthood. I did not find obvious flaws in the experimental design, lab procedures and data analysis. The manuscript is concisely written, and its conclusions are well backed up by the data. The illustrations are clear, well prepared, and are sufficient for readers to follow. Therefore, in my opinion, the submitted study certainly is novel, and high quality contribution, its findings are important and well presented, and therefore the paper fulfils the publication criteria of Nature communications. Nevertheless, I have some questions/comments/suggestions mainly to the manuscript text, and I hope that by addressing these the authors would further improve the quality of their study.

Thank you for these kind comments.

In the intro, the authors write that '... broad interest contrasts with the scarcity of literature on early life stages'. This is a very general motivation, and it implies a very exploratory and descriptive study. However, the paper is actually quite focused on a particular aspect – coordinated alteration at multiple levels following transition from vermivory to piscivory (as follows from the title). It is an appealing evolutionary question, and I would rather make it a basis of the motivation statement, rather than amorphous 'filling a gap in knowledge'.

This paragraph has been updated in response to this comment and now mentions the lack of literature even for extensively-studied species such as *Conus magus*. We introduce previous findings from Nybakken & Perron, 1988 and highlight the lack of empirical evidence for the shift from worm- to fish-hunting in this species (revised manuscript lines 46–50).

'The blind-end of the VG had differentiated into a conspicuous venom bulb (proximal end)' - I do not agree with this interpretation of proximal / distal. The term 'proximal' suggests proximity to something, and logically for a venom gland, which opens to the buccal mass, 'proximal' should denote proximity to the to the main axis of the digestive system (i.e. alimentary channel). But the blind-end of the VG (or consequently the muscular bulb) are situated at the most distanced from the buccal mass part of the venom gland, and should

therefore be considered distal. This (mis)use of proximal / distal goes back to the paper of Dutertre et al. (2014), and I would expect that in the present study the authors prefer following same terminology to be consistent. Nevertheless, for me it is simply erroneous, especially taking into consideration that the authors undertake some quality development / morphological study, and so would use morphological terms carefully.

We agree with the morphological interpretation of the reviewer. This passage has been rewritten and the rest of the text updated accordingly, replacing proximal with distal and vice-versa. The new passage now reads '[...] the VG had detached from the ventral wall of the oesophagus and only remained connected to the buccal cavity (proximal end), while its blind end had differentiated into a conspicuous venom bulb (distal end).' (revised manuscript lines 167–169).

'Morphometric analyses confirmed homology with radular teeth from vermivorous cone snails (Supplementary Fig. 3), and the presence of similar teeth in early divergent lineages such as Mitromorphidae and Borsoniidae'. - The conoidean families Mitromorphidae and Borsoniidae Mitromorphidae and Borsoniidae are not more early branching than Conidae based on the latest molecular phylogeny of the superfamily (Abdelkrim et al. 2018). Therefore, I would rather refer to them as to 'related (to cone-snails / Conidae) lineages/families of Conoidea'.

The words 'early divergent' have been removed from the text and replaced with "related conoidean lineages" as suggested. Additionally, the word 'homology' has been replaced with 'similarity' as homology cannot be inferred from morphometric analyses alone. The new passage now reads 'Morphometric analyses confirmed similarity with radular teeth from vermivorous cone snails, and the presence of similar teeth in related conoidean lineages such as Mitromorphidae and Borsoniidae suggests this trait may be plesiomorphic within the group.' (revised manuscript lines 231–235).

The sentence 'If juvenile O1-conotoxins evolved for predation on errant polychaetes, they are likely to be conserved since their first recruitment in vermivorous ancestors.' – is worded as a hypothesis and the hypothesis that does not seem to hold, based on the long branches of the juvenile O1 conotoxins on the Fig. 6. Therefore, one would expect further discussion in the subsequent sentences. However, the following sentence moves on to the adult O1 conotoxins, and seem to lack logically link with the preceding one. So, some revision is needed to close the logic gap here.

The sentence 'If juvenile O1-conotoxins evolved for predation on errant polychaetes, they are likely to be conserved since their first recruitment in vermivorous ancestors' has been deleted from the text in response to this comment.

The term 'predatory venom' at the page 10 needs some introduction/explanation.

The following sentence has been added in this section:

"[...], the milked predatory venom (MPV) was obtained from the laying female by using a fish to elicit a predatory response⁶⁰." (revised manuscript lines 430–432) and reference 60 added:

Hopkins, C. et al. A new family of *Conus* peptides targeted to the nicotinic acetylcholine receptor. *J. Biol. Chem.* **270**, 22361–22367 (1995).

The paragraph addressing the heterogeneous distribution of venom components across the vg does not seem to be well connected to the rest of the story. I think to better integrate this part, functional differences between the proximal and distal parts of the venom gland could be better placed in the ontogenetic context. If by obvious reasons differential expression of venom gene superfamilies across vg cannot be inferred for juveniles (as the gland is too small to further divide it), then some histological differences between the distal VS proximal juvenile vg can perhaps be mentioned to show the developmental basis of this heterogeneity.

This paragraph has been rewritten to address comments of both reviewers. The discussion on proximal/distal compartmentalisation of the adult VG has been removed. Unfortunately, functional characterisation of proximal and distal regions in juveniles was impeded by the small size of the juvenile VG (< 0.5mm) and this paragraph now includes the histological comparison of juvenile VS adult VG, as suggested. The new passage can be found in lines 459–468 of the revised manuscript.

Furthermore, some brief explanation would be needed for the terms like ‘excitatory’, ‘inhibitory’, ‘flaccid paralysis’, ‘rigid paralysis’, and how this information is relevant to the main story that authors tell.

In response to this comment, the following sentences have been added to the text:

“ [...] that cause hyperexcitability of axons at the venom injection site, resulting in rapid rigid paralysis of the prey⁹.” (to describe “excitatory” - revised manuscript lines 347–348)

“ inhibitory α -conotoxins [...] that target nicotinic acetylcholine receptors to induce flaccid paralysis in vertebrates^{52,53}.” (to describe “inhibitory” and “flaccid paralysis” - revised manuscript lines 354–356)

“ [...] rigid paralysis, characterised by the stiffening of the fish and continuous extension of the pectoral fins.” (to describe “rigid paralysis” - revised manuscript lines 240–242)

These sentences are supported by the following references:

9. Terlau, H. et al. Strategy for rapid immobilization of prey by a fish-hunting marine snail. *Nature* **381**, 148–151 (1996).

52. Gray, W.R. et al. Peptide toxins from *Conus geographus* venom. *J. Biol. Chem.* **256**, 4734–4740 (1981).

53. McManus, O. B., Musick, J. R. & Gonzalez, C. Peptides isolated from the venom of *Conus geographus* block neuromuscular transmission. *Neurosci. Lett.* **25**, 57–62 (1981).

In the conclusion, the authors suggest that their results potentially provide ‘a glimpse into the feeding strategy of cone snail ancestors, prior to the emergence of fish-hunting.’ This would have been a valid point, if all living cone-snail species were piscivorous. But the majority are worm-hunters, and their adult ecology/behavior/transcriptomes/proteomes allow for a more accessible inference of the feeding strategy prior to the emergence of fish-hunting.

This sentence has been deleted from the text in response to this comment.

Finally, there is a discussion context that authors ignored – comparison of the venom composition inferred from their individual with that in the previously profiled individuals of *Conus magus*. The obvious strength of the revised paper is the comparison among life history stages, that is enabled by i) controlled lab environment, and ii) closely comparable genetic background of the studied individuals resulting from the fact that they represent progeny of one female. Whereas this design allows focusing on the signal of the changes that take place during the ontogenesis, it clearly diminishes variability in the individual development. This isn't a flaw at all, as I said, but it is something that deserves being discussed. The set of questions to be addressed: 'How representative is the observed pattern for the studied species? Or across fish-hunting lineages of cone-snails?' While the present work is pioneering in breeding cone-snails in lab environment, currently there are no comparable published data sets available. There is, however, published high quality transcriptomic data on the adult venoms of *C. magus* specimens from other geographic locations (Japan, the Philippines) (Pardos-Blas et al., 2019), and I feel that by using this data for comparative purpose, authors can address variability / plasticity of venom composition among individuals / populations.

The first paragraph of the "Transcriptomics reveals venom ontogeny" section has been updated to address this comment, by discussing previous results by Pardos-Blas et al, 2019 and by the addition of a principal component analysis comparing the expression of conotoxin gene superfamilies between juvenile and adult *C. magus*. The new passage can be found in lines 271–303 of the revised manuscript.

Furthermore, in the paragraph discussing the differences between juvenile and adult M-conotoxins, a sentence referring to adult *C. magus* specimens from Japan and the Philippines has been added (revised manuscript lines 315–317).

ADDRESSING REVIEWER 2 COMMENTS:

The manuscript by Rogalski et al. interrogates the life history of *Conus magus* from eggs to metamorphosis. The research findings presented here are challenging to obtain as they rely on the ability to maintain the snails until egg laying and culture the embryos throughout hatching and metamorphosis. I think it's a beautiful study that will be well-received in the field and attract attention from related fields of inquiry.

Not only does this study expand our understanding of ontogenetic changes of the venom composition of one of the most-studied cone snail species but provides an innovative approach for the identification of new toxins that can be directly tracked to prey taxa.

We thank the reviewer for these kind comments.

Having said this, I have a few major concerns, that I would like to see addressed, one of which is the proper acknowledgement of the previous finding that *Conus magus* juveniles feed on worm and adults feed on fish. In the current version, the previous work by Nybakken and Perron is only marginally mentioned and not even discussed.

Additionally, as outlined below, all RNA-Seq raw data generated in this study have to be submitted to public repositories and accession numbers should be provided during revision.

Data presented in Figure 5 needs more rigorous, statistical analysis otherwise it is difficult to validate whether the data truly supports the claims.

Finally, some of the conclusions or hypothesis that were formulated in the paper are not well-supported by the data presented here and should be toned down. I have provided more detailed comments below.

Main Concerns:

All RNA-Seq data (and possibly also MS data) needs to be made available prior to acceptance of the manuscript. This encompasses all raw Illumina sequencing data (to be submitted to the SRA or ENA) and potentially also raw mass spec data (to be submitted to Pride or a similar public mass spec data repository). I am surprised that the authors did not already do this prior to submission.

All conotoxin precursors from our RNA-seq experiments were renamed according to the conventional conotoxin nomenclature (see methods – revised manuscript lines 613–619) and raw sequencing data have been deposited in the NCBI sequence read archive in response to this comment [SRA accession: PRJNA943605] (available upon publication of this manuscript). LC-MS mass lists have been made available as Supplementary File 2.

Either in the abstract or following the paragraph ending in line 46 it is imperative to properly cite and acknowledge the previous study on the shift from worm-hunting to fish-hunting behavior in juvenile and adult specimens of *Conus magus* (Nybakken, J. & Perron, F. E. Ontogenetic change in the radula of *Conus magus* (Gastropoda). *Mar. Biol.* 98, 239–242 (1988)). Currently, this work is not mentioned until line 204 where the main findings and hypothesis presented in that paper are not properly presented (“Predation on polychaetes was consistent with worm setae identified in the stomach content”). This prior work needs to be presented in the abstract or introduction and should also be covered in the discussion.

To address this comment, the following sentence has been added to the introduction:

“Based on dissected wild-caught specimens, *C. magus* was suggested to undergo a dietary shift from worm- to fish-hunting during ontogeny²³, but empirical evidence is lacking due to challenges accessing early life stages.” (revised manuscript lines 48–50)

Additionally, the work of Nybakken & Perron, 1988 is now discussed in the Results & Discussion section as follows: “Early vermivory in *C. magus* was previously inferred from dissected wild-caught specimens, although worm setae were only retrieved in the digestive tracts of three juveniles > 4 mm²³. Additionally, the methods used for the identification of small specimens are not mentioned and the high morphological similarity between juvenile cone snails suggests the sampling could have included other species.” (revised manuscript lines 208–213)

In the description of the juvenile radular tooth, we also added “as seen in wild-caught specimens (Nybakken and Perron, 1988)” (revised manuscript line 225).

Figure 5: there’s no easy way to assess potential significance in the data presented in Figure 5 without more information on statistical analysis in the method section and in the figure caption. Additionally, currently there is no information and no data showing how the peptides

in panel e were identified and what the scale is that was used for the heat map. The authors should provide MS/MS evidence for the identified peptides as mass-matching cannot unambiguously identify cone snail venom peptides. If this data is not available, information on the observed vs. the calculate monoisotopic mass should be provided and it should be explained how the authors know where on the HPLC run these peptides elute. Overall, the data presented in figure 5 should be more rigorously interrogated and presented.

In line with this I find the analyses presented under “Juvenile and adult *C. magus* secrete distinct VG proteomes” underwhelming. Mass matching and intact mass comparisons are really not ideal for sample comparison, especially when better methodologies (such as MS/MS sequencing) are available. Please shorten and tone down these findings or re-analyze the samples using more accurate MS-based sample comparison methods.

This figure and corresponding text have been updated in response to this comment. Statistical comparison in this study was challenged by the small sample size, with the transcriptome of 2 pooled juveniles compared to the maternal proximal and distal VG transcriptomes. However, in the paragraph ‘Transcriptomics reveals venom ontogeny’ we now compare our results on the maternal VG transcriptome with previous studies on adult *C. magus* from Japan and the Philippines to address venom variability among specimens. Furthermore, we performed a principal component analysis to compare the expression of conotoxin gene superfamilies between juvenile (n = 2, pooled) and adult (n = 2) *C. magus* (including an adult specimen from the Philippines), with the PCA biplot added in Figure 5. The data matrix, summary statistics, contribution of each variable (in per cent), PCA score and loading plots can be seen in Supplementary File 3. Our conclusions from this analysis read “Principal component analysis (PCA) confirmed the differential expression of conotoxin gene families between juvenile (n = 2) and adult (n = 2) *C. magus*. The lowest PCs (PC1 and PC2) accounted for 52.9% and 31.8% of this variation, driven mostly by overexpression of the A, T and M superfamilies in adults and overexpression of the O1, L and B2 superfamilies in juveniles.” (lines 299–303 of the revised manuscript).

Additionally, this figure now includes high-resolution MALDI-MS performed on a TIMS TOF Flex MALDI-2 (Bruker) to compare the VG proteome of juvenile (n = 20) and adult (n = 1, mother) *C. magus*. LC-MS analysis was also performed on a new ZenoTOF 7600 mass spectrometer (SCIEX) for additional comparison of masses between juvenile (n = 20) and maternal (n = 1) VG, with new MS/MS data provided to identify conotoxins from the adult VG and milked predatory venom. The section “Juvenile and adult *C. magus* secrete distinct VG proteomes” has been rewritten to incorporate this new data. The new passage can be found in lines 408–468 of the revised manuscript. Heatmaps built on the 15 dominant conotoxins detected in our MALDI- and LC-MS experiments are available as Supplementary Figure 8.

Minor concerns/comments:

Lines 31-32: Do all cone snails have a planktonic stage? I thought that some transition into a benthic stage without a prior planktonic stage.

This sentence has been modified as follows:

“This group of predatory gastropods has evolved within a biphasic lifecycle, with most species hatching as free-swimming larvae that become benthic carnivorous juveniles after metamorphosis.” (revised manuscript lines 30–32)

Additionally, the figure 1 legend has been modified from “Cone snails have a biphasic life history that includes a planktonic larva that becomes a benthic carnivorous juvenile after metamorphosis.” to “*Conus magus* has a biphasic life history that includes a planktonic larva that becomes a benthic carnivorous juvenile after metamorphosis.” (revised manuscript lines 90–91)

Lines 32-34 Clarify that not all cone snails have harpoon-like radula teeth.

This sentence has been modified in response to this comment and now reads as follows: “Predatory feeding after metamorphosis relies on the deployment of potent neurotoxins (conotoxins) secreted in a long tubular venom gland and injected via highly modified, hollow radular teeth.” (revised manuscript lines 32–34)

Line 9: Cite Louise Page 2012 after “the foregut...”.

The work of Page (2012) has been cited in the sentence:

‘The venom apparatus of marine cone snails (Gastropoda: Conidae) is an example of evolutionary innovation that evolved through morphological modifications of the foregut⁵’ (revised manuscript lines 27–29).

5. Page, L. R. Developmental modularity and phenotypic novelty within a biphasic life cycle: morphogenesis of a cone snail venom gland. *Proc. Biol. Sci.* **279**, 77–83 (2012).

Line 36: these are more accurate references for research on the shift in prey type: <https://pubmed.ncbi.nlm.nih.gov/24878223/> and <https://www.sciencedirect.com/science/article/abs/pii/S0024406601905449>

The references have been updated following reviewer 2 suggestions:

‘This sophisticated feeding strategy has allowed these slow-moving predators to initially feed on worms, and more recently facilitated the evolutionary shift to mollusc- and fish-hunting^{11,12}.’ (revised manuscript lines 34–36).

11. Puillandre, N., P. et al. Molecular phylogeny and evolution of the cone snails (Gastropoda, Conoidea). *Mol. Phylogenet. Evol.* **78**, 290–303 (2014).

12. Duda Jr, T.F., Kohn, A.J. & Palumbi, S.R. Origins of diverse feeding ecologies within *Conus*, a genus of venomous marine gastropods. *Biol. J. Linn. Soc.* **73**, 391–409 (2001).

Line 56: fix typo (“un”) and clarify that these toxins could have also been retrieved by exon capture or genome sequencing (e.g., an untapped source of novel bioactive venom peptides that would otherwise only be accessible through exon capture or genome sequencing).

The word “un” has been replaced by “an” and the sentence has been modified according to reviewer 2 suggestion to read as follows:

'Our results [...] highlight the potential of juvenile cone snails as an untapped source of novel bioactive venom peptides that would otherwise only be accessible through exon capture or genome sequencing.' (revised manuscript lines 58–61).

Line 62: clarify how the species and sex was identified here (many snails are hermaphrodites and this should be clarified here).

In response to this comment, the following sentence has been added at the beginning of this paragraph:

"Like the vast majority of caenogastropods, cone snails are gonochoric (males and females exist as separate individuals)." (revised manuscript lines 67–68)

Figs 1-3 and Supporting Figs 1-2 are beautiful and very informative presentations of the *C. magus* life history. Wonderful work!

The authors thank reviewer 2 for these kind comments.

Lines 220-22: This is a very intriguing hypothesis, and I am looking forward to seeing the findings on juvenile mollusk hunters.

Transcriptome analysis: it is not clear from the results or methods section how many individual adult venom glands were sequenced. This is important to mention and discuss as there can be significant intraspecies variation in venom gene expression. I think the data presented here goes beyond what would be expected for intraspecies variation, but nevertheless, it's important to address this issue with scientific rigor.

In response to this comment, the first sentence of the "Transcriptomics reveals ontogeny" section has been rewritten as:

"To investigate the relationship between prey-preference and venom biochemistry, transcriptomes were generated from embryonic (n ~ 500), juvenile (n = 2) and adult (n = 1) *C. magus*." (revised manuscript lines 271–273). This information can also be found in the methods section ("RNA extraction and sequencing") (revised manuscript lines 562–563).

To address intraspecific venom variation, the work of Pardos-Blas, 2019 has been discussed in the "Transcriptomics reveals venom ontogeny" section. The new passage reads as follows: 'These findings were consistent with recent studies where the M, O1, T and A superfamilies dominated the VG transcriptomes of adult *C. magus* specimens from Japan and the Philippines³⁸, suggesting different populations of this species share a similar repertoire of conotoxins at adulthood.' (revised manuscript lines 291–294).

38. Pardos-Blas, J. R., Irisarri, I., Abalde, S., Tenorio, M. J. & Zardoya, R. Conotoxin diversity in the venom gland transcriptome of the Magician's Cone, *Pionoconus magus*. *Mar. Drugs* **17**, 10 (2019).

Additionally, a principal component analysis was performed to compare the expression of conotoxin genes between juvenile (n = 2) and adult (n = 2) *C. magus* (including published data for a specimen from the Philippines). (revised manuscript lines 299–303).

Furthermore, in the section discussing the difference between juvenile and adult M conotoxins, the following sentence has been added:

“This precursor was also found in adult *C. magus* from Japan and the Philippines, along with two other precursors encoding the same mature peptide.” (revised manuscript lines 315–317)

Lines 278-286: very nice finding on the differences in the M-superfamily toxins

Lines 320 - 322: I disagree that the low expression of alpha conotoxins in the predatory venom suggests that there are used in defense. There would be several alternative hypotheses, such as prey-specific use of alpha conotoxins or a lack of reproducing natural predatory behavior in the lab. Either remove this sentence or provide potential alternative explanations.

The previous sentence has been deleted from the text in response to this comment.

Lines 330-331: It's not clear to me what study is being referred to here. Provide reference or clarify whether there's any data showing that these toxins are actually injected into worms.

This passage has been rewritten as “Whether juvenile κ A- and δ -conotoxins may be used to deter vertebrate predators and/or to facilitate predation on worms remains to be determined.” (revised manuscript lines 364–366)

Line 359: Provide confidence values for alphafold structures.

During revision of the manuscript, the authors decided to remove the AlphaFold structures (replaced by a PCA biplot in Figure 5).

Lines 411-414: The differences in the data presented for the distal vs the proximal VG are weak and any conclusions from this data should be removed. There may be larger differences in other species but, it does not seem that this is the case here.

While our transcriptomic and proteomic data suggests that the expression of conotoxins is heterogenous along the adult VG, we agree that there is no obvious proximal/distal compartmentalisation as shown previously for other species (e.g. Dutertre et al. 2014) and the last paragraph was rewritten accordingly. The new passage can be found in lines 459–468 of the revised manuscript.

Several previous studies have provided phylogenetic evidence supporting that ancestral *Conus* were worm hunters. Thus, I strongly suggest changing the text from “support the hypothesis...” to “support previous evidence...”. These papers should be cited here: <https://pubmed.ncbi.nlm.nih.gov/24878223/> and <https://www.sciencedirect.com/science/article/abs/pii/S0024406601905449>

This passage has been modified according to reviewer 2 suggestions, with the suggested references added. This section now reads as follows:

“For the first time, we characterised the venom arsenal of juvenile cone snails. Our findings support previous evidence that fish-hunters evolved from vermivorous ancestors^{11,12}.” (lines 474–476).

11. Puillandre, N., P. et al. Molecular phylogeny and evolution of the cone snails (Gastropoda, Conoidea). *Mol. Phylogenet. Evol.* **78**, 290–303 (2014).

12. Duda Jr, T.F., Kohn, A.J. & Palumbi, S.R. Origins of diverse feeding ecologies within *Conus*, a genus of venomous marine gastropods. *Biol. J. Linn. Soc.* **73**, 391–409 (2001).

I love the supporting video of the juvenile (393262_0_video_6976165_rjkpj4.mp4)! Is it possible to label

The Supplementary Movie 2 is labelled (Syllid, rostrum, foot) in the first 4 seconds of the video.

REVIEWERS' COMMENTS

Reviewer #1 (Remarks to the Author):

I am happy with the revisions made by the authors to satisfy my previous comments. I have only few further questions:

1. Sentence at lines 366-370: '...but lacked the -ECCS- motif where the glutamic acid is post-translationally modified to a γ -carboxyglutamate' – I don't see this motif in the reference sequence from *Conus pennaceus* on the Supp. Figure 6.
2. Line 436 – please, identify which sister species is/are ment.
3. Line 442 – same as previous.

In my opinion, the paper is otherwise suitable for publication.

Reviewer #2 (Remarks to the Author):

This study by Rogalski et al. beautifully describes the changes in the life history of *Conus magus* from juveniles to adults. I appreciate that the authors addressed my previous concerns, particularly regarding the availability of RNA-Seq datasets and the limitations of the data shown in Figure 5. I fully understand that it is difficult to obtain sufficient individuals for such a study and including a comparative analysis to previously published venom gland datasets of adult *Conus magus*, as suggested by the other reviewer, provided more confidence for the differences in the venom composition between juveniles and adults. This also highlights the importance of making raw data available to other researchers in the community. I have no further comments and fully support publishing this paper that will surely be well received by the *Conus* venom field and beyond.

RESPONSE TO REVIEWERS' COMMENTS

ADDRESSING REVIEWER 1 COMMENTS:

I am happy with the revisions made by the authors to satisfy my previous comments. I have only few further questions:

1. Sentence at lines 366-370: '...but lacked the –ECCS– motif where the glutamic acid is post-translationally modified to a γ -carboxyglutamate' – I don't see this motif in the reference sequence from *Conus pennaceus* on the Supp. Figure 6.

The motif –ECCS– can be seen in the reference sequence from *C. pennaceus* (residues 59–62) in Supplementary Figure 6.

2. Line 436 – please, identify which sister species is/are ment.

This sentence has been modified in response to this comment and now reads as follows: "This mass range and elution window are typical of excitatory O-glycosylated κ A-conotoxins, which dominate the predatory venom of sister species *C. catus* and *C. striatus*." (revised manuscript lines 332–334)

3. Line 442 – same as previous.

This sentence has been modified in response to this comment and now reads as follows: "In contrast, the MPV lacked the more hydrophobic venom components and the smaller molecular weight peptides present in either region of the adult VG, as previously observed in the piscivorous *C. consors*." (revised manuscript lines 338–341)

In my opinion, the paper is otherwise suitable for publication.

ADDRESSING REVIEWER 2 COMMENTS:

This study by Rogalski et al. beautifully describes the changes in the life history of *Conus magus* from juveniles to adults. I appreciate that the authors addressed my previous concerns, particularly regarding the availability of RNA-Seq datasets and the limitations of the data shown in Figure 5. I fully understand that it is difficult to obtain sufficient individuals for such a study and including a comparative analysis to previously published venom gland datasets of adult *Conus magus*, as suggested by the other reviewer, provided more confidence for the differences in the venom composition between juveniles and adults. This also highlights the importance of making raw data available to other researchers in the community. I have no further comments and fully support publishing this paper that will surely be well received by the *Conus* venom field and beyond.